# *IL17F* Expression as an Early Sign of Oxidative Stress-Induced Cytotoxicity/Apoptosis

**DOI:** 10.3390/genes13101739

**Published:** 2022-09-27

**Authors:** Mario Bauer, Beate Fink, Ulf Anderegg, Stefan Röder, Ana Claudia Zenclussen

**Affiliations:** 1Department of Environmental Immunology, Helmholtz Centre for Environmental Research-UFZ, 04318 Leipzig, Germany; 2Department of Dermatology, Venereology and Allergology, Leipzig University, 04103 Leipzig, Germany

**Keywords:** L17F, apoptosis, BEAS-2B, THP-1, fibrosis, primary dermal fibroblasts

## Abstract

Interleukin 17F (IL17F) has been found to be involved in various inflammatory pathologies and has recently become a target for therapeutic purposes. In contrast to IL17F secreted by immune cells, the focus of this study is to describe the triggers of IL17F release in non-immune cells with a particular focus on IL17F-induced fibrosis. IL17F induction was examined in human lung epithelial (BEAS-2B) and myeloid cell lines as well as in peripheral blood mononuclear cells after in vitro exposure to aqueous cigarette smoke extract (CSE), inorganic mercury, cadmium or the apoptosis inducer brefeldin A. Fibrosis was examined in vitro, evaluating the transition of human primary dermal fibroblasts to myofibroblasts. We observed that all stressors were able to induce IL17F gene expression regardless of cell type. Interestingly, its induction was associated with cytotoxic/apoptotic signs. Inhibiting oxidative stress by N-acetylcysteine abrogated CSE-induced cytotoxic and *IL17F*-inducing effects. The induction of *IL17F* was accompanied by IL17F protein expression. The transition of fibroblasts into myofibroblasts was not influenced by either recombinant IL17F or supernatants of CSE-exposed BEAS-2B. In addition to IL17F secretion by specialized or activated immune cells, we underscored the cell type-independent induction of IL17F by mechanisms of inhibitable oxidative stress-induced cytotoxicity. However, IL17F was not involved in dermal fibrosis under the conditions used in this study.

## 1. Introduction

Interleukin 17F (IL17F) [1] belongs to the IL17 family of cytokines, which consists of the six members IL17A, -B, -C, -D, -E and -F. Expression of IL17F is not restricted to a single cell type. In addition to specific T helper (Th)17 cells [2], it is produced in other activated immune cells such as mast cells, basophils and epithelial cells [3,4]. It can act as a heterodimer with IL17A or as a homodimer via an IL17RA/RC or IL17RC/RC receptor complex, respectively, to induce inflammatory cytokines, chemokines and metalloproteinases via several signaling pathways [5]. IL17F is a pro-inflammatory cytokine involved in autoimmune and inflammatory diseases [3,6].

Under certain inflammatory conditions, the expression of IL17F is somewhat increased, irrespective of tissue type. Higher levels of IL17F have been found in the sputum of patients with cystic fibrosis (CF) undergoing pulmonary exacerbation [7], in saliva of patients with periodontitis [8], in psoriatic skin lesions [9] or in sera of patients with atopic asthma or systemic sclerosis [10,11]. The cellular origin of IL17F under these pathological conditions is rather unknown.

For tissues with a barrier function, such as skin or intestine, there is a strong dichotomy of IL17 effects. Outcomes of IL17 signaling can be beneficial or detrimental depending on the micromilieu of the responding cells [12]. Under normal conditions, IL17 is more responsible for host defense. By contrast, in autoimmune diseases, IL17 contributes to pathogenic inflammation [13]. Currently, IL17F is used as a target for clinical intervention. Dual inhibitors and bispecific antibodies simultaneously targeting IL17A and IL17F are under investigation in different clinical studies to treat psoriasis and psoriatic arthritis [14], or to prevent pathological bone formation, which may be helpful in the treatment of ankylosing spondylitis [15].

In addition to IL17F secreted by activated immune cells, we here describe the close relationship between strong IL17F induction and cytotoxic/apoptotic stress in vitro. Cellular oxidative stress was induced in three different human cell types by different xenobiotics such as the composed mixture of aqueous cigarette smoke extract (CSE), the metals cadmium and mercury as well as the apoptosis inducer brefeldin A. Since the majority of patients (up to 70%) suffering from idiopathic pulmonary fibrosis smoked in early life [16] and elevated serum concentration of IL17F has been shown to correlate with the severity of fibrosis-causing autoimmune diseases [11], we sought to examine whether IL17F also plays a role in the induction of fibrosis under experimental in vitro conditions. 

## 2. Materials and Methods

### 2.1. Chemicals Used for Exposure Experiments

For histochemical staining, monensin from Sigma-Aldrich (Munich, Germany) and Hoechst 33,342 from Invitrogen (Carlsbad, CA, USA) were used. For exposure experiments, cadmium chloride, mercury chloride, brefeldin A (17.8 µM) and N-acetylcysteine (NAC) were used from Sigma-Aldrich (Munich, Germany). Annexin V-FITC was purchased from Biolegend (Koblenz, Germany) and used as recommended by the supplier.

Cigarette smoke extract (CSE) was freshly prepared according to the protocol described by Adenuga and coworkers [17]. Briefly, research-grade reference cigarettes (1R6F) from the University of Kentucky (Tobacco Health Research, Lexington, KY, USA) were used to prepare CSE by bubbling smoke from one cigarette into 2 mL of RPMI 1640 without supplements at a rate of 1 cigarette/minute. Afterwards CSE was sterile-filtered through a 0.22 μm filter (Sartorius, Göttingen, Germany). The control was obtained by using air instead of cigarette smoke by the same procedure. 

The recombinant cytokines rIL17F and rTGF-β1 were purchased from Peprotech (Biozol, Eching, Germany).

### 2.2. Cell Culture Experiments

Human adherent lung epithelial cell line, BEAS-2B, and human monocytic leukemia cell line, THP-1, were obtained from American Type Culture Collection (ATCC, Wesel, Germany). Peripheral blood mononuclear cells (PBMC) were isolated from blood of three pseudonymous healthy volunteers obtained from the blood bank at the University of Leipzig. PBMCs were obtained by density-gradient centrifugation using Ficoll-Paque (GE Healthcare, Solingen, Germany). For the use of human blood samples, the study received approval from the Ethics Committee of the University of Leipzig (reference number 079-15-09032015). Cells were kept in RPMI-1640 medium supplemented with 10% fetal bovine serum (FBS), 0.05 mM 2-mercaptoethanol, 100 units/mL penicillin and 100 µg/mL streptomycin, and cultured at 37 °C under 5% CO_2_ in a humidified atmosphere. THP-1 cells growing in suspension were seeded for 2 days in the presence of 10 ng/mL phorbol-12-myristat-13-acetat (PMA, Sigma-Aldrich, Munich, Germany) to obtain adherence. Afterwards, cells were kept in medium without PMA for another 24 h before exposure experiments started. 

In vitro experiments to analyze fibrosis were performed on primary human dermal fibroblasts (dFb) isolated from healthy breast skin as previously described [18]. Experiments were approved by the local ethics committee (065-2009) and conducted according to the Declaration of Helsinki principles (1975). Cells were cultured with Dulbecco’s Modified Eagle Medium (DMEM, anprotec, Bruckberg, Germany) supplemented with 10% FBS (PAN-Biotech, Aidenbach, Germany) and 1% penicillin/streptomycin (anprotec) at 37 °C, 5% CO_2_ until confluence. Exposure of dFb was performed between passages 2–4. In comparison to TGF-β1 (5 ng/mL) as an inducer of transition of fibroblasts to myofibroblasts (FMT, induction of fibrosis), dFb was exposed to IL17F (100 ng/mL) and supernatants of CSE-exposed BEAS-2B. Supernatants of BEAS-2B were acquired after exposure to CSE for 2 h, and subsequent incubation for another 24 h in normal media. 

### 2.3. Cell Viability Assays

Cell viability was measured by two methods using intracellular (MTT-assay) or extracellular (G6PD release) for signs of cell damage. The MTT assay based on the intracellular conversion of MTT (3-(4,5-dimethylthiazol-2-yl)-2,5-diphenyltetrazolium bromide, Applichem, Darmstadt, Germany) to formazan crystals by mitochondrial dehydrogenases [19]. After exposure, 30 µL MTT solution (5 mg/mL MTT in PBS) was added to 150 µL culture medium and incubated for 3 h at 37 °C. To dissolve the formazan crystals, 150 µL of stop solution (20% sodium dodecyl sulfate in 50% N,N-dimethylformamid, pH 4.7) was added and incubated at 37 °C overnight. Absorbance was measured at λ = 570 nm using a microplate reader (TECAN, Salzburg, Austria). 

Plasma membrane damage was measured by the release of cytosolic glucose 6-phosphate dehydrogenase (G6PD) into medium with the Vybrant^TM^ Cytotoxicity Assay Kit (Molecular Probes, Eugene, OR, USA), according to manufacturer’s instructions. The morphological changes of adherent epithelial cells as a part of transition from polygonal toward round cell shape (illustrated in Appendix A) were estimated by quantification of cell-free area using hybrid cell count option for phase contrast pictures of the Fluorescence Microscope BZ-X800 (Keyence, Neu-Isenburg, Germany). 

### 2.4. Gene Expression by Quantitative PCR

Total RNA of samples (in triplicates) was prepared by using TRIzol reagent (ThermoFisher Scientific, Göttingen, Germany) according to the manufacturer’s instructions. The cDNA synthesis was carried out with 1 µg of RNA by using the ImProm-II^TM^ Reverse Transcription System (Promega, Mannheim, Germany). Intron-spanning primers were designed and appropriate UPL probes (Appendix A) were selected by the Universal Probe Library Assay Design Center (http://qpcr.probefinder.com/organism.jsp (accessed until 31 December 2019)). Comprehensive gene expression was measured using the 96.96 or 48.48 Dynamic Array (Fluidigm, San Francisco, CA, USA). Genes associated to fibrosis were measured using the Roche Lightcycler 480 (Roche, Mannheim, Germany). Gene expression values were determined by using the 2^−∆∆CT^ method [20]. 

### 2.5. Immunohistochemistry

Adherent cells were plated and stained in 96-well plates. Secretion of IL17F into culture medium was blocked with 7.5 µM monensin applied during the last 3 h of exposure. After medium withdrawal, cells were fixed in 4% paraformaldehyde in phosphate buffer saline (PBS) for 30 min at 37 °C. After two-fold wash steps with PBS/0.1%Tween-20 (PBS/Tween), cells were blocked with blocking solution (0.3 M glycine, 0.3% bovine serum albumin, 10% goat serum) for 30 min at room temperature following incubation with anti-human IL17F (1:100, R&D, Wiesbaden (Nordenstadt), Germany) or the mouse IgG2b isotype control (Dako, Hamburg, Germany) in blocking solution at 4 °C overnight. After two-fold wash steps with PBS/Tween, goat-anti-mouse IgG2b-FITC (1:500, Southern Biotech/Biozol, Eching, Germany) was applied for 1 h at room temperature in dark. For nuclear staining, Hoechst 33,342 (1:2000, Invitrogen, Carlsbad, CA, USA) was used. Images for controls are depicted in Appendix A. Images were acquired with the All-in One Fluorescence Microscope BZ-X800 (Keyence, Neu-Isenburg, Germany).

### 2.6. Statistical Analysis

Significant differences were analyzed with analysis of variance (ANOVA). Test of normality was performed using the Shapiro–Wilks test. All *p*-values < 0.05 were considered to be statistically significant. For multiple testing, the Bonferroni correction was considered. If not otherwise indicated, statistical calculations were performed with Statistica for Windows version 10 (StatSoft Inc. Europe, Hamburg, Germany). Heatmaps of correlation analysis and principal component analysis (PCA) were performed in R framework (R3.6.1, https://cran.r-project.org (accessed on 30 June 2021)) for Windows using the open-source solution for statistical computing, the SINGuLAR Analysis Toolset (Fluidigm, San Francisco, CA, USA). 

## 3. Results

### 3.1. Induction of IL17F Gene Expression by CSE in Different Cell Types

Pilot experiments analyzing the effect of different stressors on immune cell-related gene expression using the myeloid cell line THP-1 indicated a specific and strong induction of *IL17F* when exposed to aqueous cigarette smoke extract (CSE, Appendix A). To show that this induction of *IL17F* by CSE was not dependent on cell type, we analyzed the effect of CSE on three different human cell types. We investigated *IL17F* expression together with five other members of the IL17 family (*IL17A, -B, -C, -D* and *-E*) and three IL17 receptors (*IL17RA*, *-RB* and *RC*). In the lung epithelial cell line BEAS-2B, *IL17F* was by far the most induced gene (about 120-fold at 4% CSE compared to the control, *p* < 0.001) at an early timepoint of 4 h, in contrast to the majority of the other IL17-related genes (Figure 1A). Expression was further increased after 24 h (up to 770-fold, *p* < 0.001, Table 1, Figure 1A). Similarly, in myeloid THP-1 (Figure 1B) and blood PBMCs (Figure 1C), *IL17F* was by far the gene that was most induced by CSE exposure for 24 h (about 290- and 200-fold, respectively).

With respect to the adverse effect of CSE on viability at higher concentrations, it became evident that *IL17F* induction was related to cytotoxicity (Figure 2). Although cell membrane integrity (Figure 2A) was not affected, depressed viability (EC75, MTT assay) was reached with 4% CSE (Figure 2B). Additionally, at high concentrations of CSE, epithelial BEAS-2B lost their polygonal shape and became irreversibly more rounded (Figure 2C, Appendix A).

### 3.2. Effect of NAC on CSE-Induced Gene Expression

To determine whether reactive oxygen species (ROS) are involved in early CSE-induced *IL17F*, the epithelial cell line BEAS-2B was co-treated with CSE and NAC for 4 h. NAC strongly blocked the early effect of CSE on *IL17F* (Figure 3). By contrast, the early CSE-induced *HMOX1* (NRF2-mediated) and *CYP1A1* (AhR-mediated) was not eliminated by NAC; rather, the reduction in induction at high CSE concentration for *HMOX1* was abrogated. The cell-protecting effect of NAC against oxidative stress was apparent exclusively in the concomitant presence of NAC and CSE and was microscopically visible by the absent transition from polygonal to round cells at higher CSE concentrations (Appendix A). The effect of NAC also remained during a longer exposure up to 24 h (Appendix A).

### 3.3. Relationship between IL17F Induction and Cytotoxicity

Because of the effect of NAC and the close relationship between *IL17F* induction and cytotoxicity, we explored the effect of further inducers of oxidative stress, in particular cadmium (Cd^2+^, Figure 4A) and mercury (Hg^2+^, Figure 4B). The cytotoxic effect of both components on BEAS-2B after 24 h exposure was accompanied by loss of polygonal shape (shown for Hg^2+^ in Appendix A). Regardless of the different strengths of the cytotoxic effect (MTT assay), *IL17F* was induced at cytotoxic concentrations for both Cd^2+^ and Hg^2+^, with the highest *IL17F* expression at the concentration at which BEAS-2B completely lost its polygonal shape. For Hg^2+^, the expression pattern of others included IL17 genes was similar to CSE exposure (Figure 4C).

### 3.4. IL17F Protein Induction by CSE and Brefeldin A

To verify whether IL17F gene expression is translated into protein release and could be induced in cells that are undergoing apoptosis, IL17F protein expression was examined by immune fluorescence after treatment with the apoptosis inducer brefeldin A in comparison to CSE in BEAS-2B (Figure 5). In contrast to an exposure of 4 h, IL17F protein expression was apparent after 24 h by a cytoplasmic granular pattern for both brefeldin A and CSE. Concomitantly, the frequency of cells positive for the apoptosis stain Annexin V was increased (Appendix A), indicating the involvement of apoptosis in the cytotoxic effect of CSE. Figure 6 schematically illustrates the cell type-independent induction of IL17F under oxidative stress-induced cytotoxic/apoptotic conditions in vitro, compared to IL17F secretion by specialized Th17 immune cells.

### 3.5. Comprehensive Analysis of Exposure-Induced Gene Expression 

To gain more insight into the mechanism of exposure-induced IL17F, we analyzed transcripts of genes related to stress, apoptosis and inflammation and those that are involved in IL17F transcript release (Appendix A). By comparing 39 investigated genes, it became evident that *IL17F* was the only gene showing strong induction independent of cell type and stressor (about 15 up to 700-fold). In a similar manner, *DDIT3*, a marker of ER stress, *HMOX1*, indicating oxidative stress, and *CDKN1A*, related to senescence, were induced, but in contrast to *IL17F*, this occurred at lower concentrations of stressors and, in some cases, at a much lower degree of induction. The best correlation for *IL17F* was found to be *IL17B* in CSE-induced BEAS-2B (Appendix A). A heatmap of the correlation analysis shows a strong increase in most of the genes as CSE concentration increases. Concentration-dependent stratification was confirmed by principal component analysis (PCA, Appendix A). Due to the inhibition of protein synthesis by brefeldin A, the pattern of its induced gene expression differed from that of CSE and metals. However, a pronounced dose-dependent transcript induction was present for the stress-indicating heat shock proteins *HSPA1A* and *HSPA2*, pro-inflammatory *TNF* and *IL22* and pro-apoptotic *FASLG*. Gene expression of proteins involved in IL17F transcript regulation such as *RORC*, *SOCS3* and *IL23A* was inconspicuous (Appendix A). Furthermore, no stratification to the functionality of genes was apparent.

### 3.6. Influence of IL-17F on In Vitro Fibrosis Using Primary Dermal Fibroblasts

To verify a potential involvement of CSE-induced *IL17F* in fibrosis, we examined in vitro the transition of fibroblasts to myofibroblasts (FMT) using primary human dermal fibroblasts. Although recombinant IL17F (rIL17F) and the inducer of FMT TGF- β1 (rTGF-β1), both induced *IL6* in fibroblasts (Appendix A), neither rIL17F alone nor supernatants from lung epithelial cell line BEAS-2B, exposed to CSE, induced FMT (Figure 7). The FMT was evaluated by increased gene expression of the four FMT-associated genes α smooth muscle actin (*ACTA2*), collagen type I (*COL1A1*), palladin (*PALLD*) and ED-A fibronectin (*FN1*). In contrast to rIL17F and supernatants from CSE-exposed BEAS-2B, rTGF-β1 induced FMT independently of the presence of supernatants from CSE-exposed BEAS-2B.

## 4. Discussion

This study aims to understand the ability of CSE to induce IL17F production and whether this is associated with fibrosis. This idea arose from results of preliminary experiments using THP-1 cells that showed a strong and seemingly highly specific induction of *IL17F* by CSE, compared to five other stimulants and their combinations and compared to 47 other genes. In contrast to IL17F induction in specific immune cells by chemokines, the CSE-induced *IL17F* seems to rely on another mechanism of IL17F induction, which we wanted to investigate in more detail. In order to exclude a cell-type specific effect, we replicated experiments on different human cell types (epithelial BEAS-2B, myeloid THP-1, and primary PBMC). In order to exclude CSE-specific effects, we included inorganic mercury, cadmium and the apoptosis inducer brefeldin A. 

First of all we compared exposure-induced *IL17F* induction to other members of the IL17 family as well as reported genes related to be involved in IL17 induction. The exposure-induced *IL17F* induction was unique, correlating neither with other IL17 members such as *IL17A,* which can form heterodimers, nor with that of IL17 receptors *IL17RA* or *RC,* which can bind IL17F as a heterodimer or homodimer, respectively [22]. Interestingly *IL22*, a further Th17 effector cytokine, was concomitantly induced under these cytotoxic/apoptotic conditions, assuming a common transcriptional activation for both cytokines. Since genes associated with IL17-inducing pathways such as *RORC, IL23/STAT3, SOCS3* and *IL6/RORA* [23,24] remained unchanged, an IL17F-inducing mechanism specific to cytotoxic/apoptotic conditions is suggested. The expression of other members of the IL17 family under these conditions differed depending on the cell type. For example, in contrast to the myeloid THP-1 cell line, in the lung epithelial BEAS-2B cell line, *IL17B* correlated well with *IL17F*, possibly for its role in neutrophil attraction under stress conditions [25]. 

CSE-induced IL17F expression was associated with cytotoxicity/apoptosis. To discover the reason for the chemical-induced cytotoxic/apoptotic signs, we hypothesized that oxidative stress could play a decisive role. Thus, we examined (i) results after using inducers of oxidative stress, (ii) a marker gene indicating oxidative stress and (iii) the effects of an inhibitor of oxidative stress. As inducers of oxidative stress, we used cadmium (Cd) and mercury (Hg), metals which are also found in cigarette smoke. Cadmium is potent to induce oxidative stress, although by itself it is not redox-active. It is unable to generate reactive oxygen species (ROS) directly [26]. In contrast, Hg-induced oxidative stress may occur due to both prooxidant action of the metal and a decrease in antioxidant enzymes. Despite the absence of direct indications, it can be proposed that mercury may induce endoplasmic reticulum (ER) stress [27]. Indeed, both metals induced *IL17F*. ER stress could be also involved in cytotoxic mechanisms, since transcripts of the ER stress marker DNA damage inducible transcript 3 (*DDIT3*), were strongly induced. 

With respect to CSE, we propose that antioxidant proteins such as HMOX1, becoming activated via antioxidant response elements by a nuclear factor erythroid 2-related factor 2 (NRF2), should be induced. Indeed, *HMOX1* was augmented upon CSE exposure but its induction could not be reduced by NAC. Therefore, it can be assumed that despite the cytoprotective effect of NAC, CSE-exposed cells are still under stress, and HMOX1 exerts its protective effect via heat shock proteins or anti-inflammatory properties [28]. In addition, the cytoprotective effect of NAC is caused less by ROS scavenging and more by intracellular generation of protein-protecting sulfane sulfur species from NAC [29]. Besides the lack of effect of NAC on CSE-induced *HMOX1* expression, other CSE-induced genes remained unchanged. Among them was *CYP1A1,* a gene induced by the aryl hydrocarbon receptor (AhR). Since AhR agonists also induced oxidative stress via the formation of ROS by AhR-induced cytochrome P450 (CYP) enzymes [30], cytoprotection through NAC could also rely on the inhibition of AhR-triggered ROS. However, neither AhR antagonist (α-naphtoflavone) nor AhR agonist (benz(a)pyrene) impaired CSE-induced *IL17F* (Appendix A). Thus, ROS induction via an AhR-dependent pathway does not seem to be responsible for the IL17F induction. Nevertheless, (i) the effects of metals, (ii) the induction of *HMOX*, (iii) the effect of NAC, all together, give rise to assume a general role of oxidative stress in induction of IL17F. 

The presence of apoptosis signs elicited by CSE both at the gene expression level and in histological Annexin V staining suggests a prominent role of apoptosis in IL17F induction. Similar apoptotic signs have also been described in airway epithelial cells from cigarette smoking-induced COPD patients [31]. The impact of apoptosis on IL17F was further substantiated by brefeldin A-inducing *IL17F* as well as protein expression. Brefeldin A as an endoplasmic reticulum (ER)-Golgi transport inhibitor causes accumulation of proteins in the ER, thus causing ER stress accompanied by the formation of reactive oxygen species and depletion of GSH, subsequently causing apoptosis [32]. Because of its inhibition of protein synthesis, it influences transcription of genes. Thus, the brefeldin A-induced gene expression pattern did not fit well with that of CSE or metals even when newly formed proteins would impair transcription. Nevertheless, *IL17F*, *DDIT3,* and the pro-apoptotic *FASLG* were inducible. This indicated that transcription of these genes is less under control of exposure-triggered newly-synthesized proteins, which might explain the early transcriptional activation of IL17F. 

The putative role of IL17F in apoptosis induction remains speculative. We have not examined this in detail. However, since IL17-secreting Th17 cells are resistant to glucocorticoid-induced apoptosis [33], an apoptosis-promoting effect of IL17F seems rather unlikely. A somewhat anti-apoptotic effect was shown by the inhibition of Fas-induced cell death in Fas-sensitive T cells exposed to recombinant IL17F [34].

With the examined cytotoxicity/apoptosis-inducing chemicals we could confirm reports of IL17F induction by similar acting agents. Indeed, this has been reported for the African swine fever virus (ASFV) in macrophages, for UV-A in fibroblasts and for apigenin in pancreatic cancer cells. In swine macrophages, the virus upregulated *IL17F* and cytokines of the TNF family including *FASLG* and *TNF* [35]. In skin fibroblasts, in a similar manner UV-A induced IL17F and proteins involved in apoptosis [36]. Interestingly, similar to the apoptosis-preventing effect of NAC, the effect of UV-A was restored by the antioxidant rutin. Furthermore, in pancreatic cancer cells, the citrus fruit bioactive flavonoid apigenin strongly upregulated IL17F (>100-fold) and induced cell death through inhibition of the glycogen synthase kinase-3β (GSK3β)/nuclear factor kappa B (NFKB) signaling pathway [37]. Recently it was suggested that CSE is also able to inactivate GSK-3β via activation of the PI3K/Akt pathway [38]. The question of whether GSK-3β has a central role in apoptosis remains elusive, since deactivation of GSK-3β in cardiomyocytes protected against doxorubicin-induced apoptosis [39]. 

To clarify whether IL17F release from cells undergoing cytotoxicity/apoptosis can cause fibrosis, we analyzed FMT in human primary dermal fibroblasts. This is used as an indicator of fibrosis. However, we could not detect pro-fibrotic effects of CSE, either of rIL17F or of supernatants from CSE-exposed epithelial cells in which apoptosis-related IL17F was induced. Due to easier availability, dermal fibroblasts were used primarily assuming similar conditions to lung fibroblasts with respect to IL17RA. A functional IL17RA is essential for IL17A-induced FMT in lung fibroblasts [40] and IL17RA is also found together with IL17RC on dermal fibroblasts [41]. According to Levy et al. [42], we confirmed the functionality of IL17RA by induction of IL-6 by IL17F in dermal fibroblasts. However, since we focused solely on dermal fibroblasts, we cannot exclude that IL17F-induced fibrotic effects may occur in other tissue-specific fibroblast types. The putative role of cytotoxicity/apoptosis-associated induction of IL17F could, therefore, not be definitively clarified in our experiments.

## 5. Conclusions

In addition to IL17F secretion by specialized or activated immune cells under physiological or pathological conditions, we have now revealed a cell type-independent induction of IL17F under oxidative stress-induced cytotoxicity in vitro. However, this IL17F does not appear to play a role in dermal fibrosis. Because of its strong induction and association with cytotoxicity/apoptosis, it could be of interest to establish whether IL17F could serve as a diagnostic marker to follow therapeutic interventions or whether its cellular origin has an impact on therapeutic interventions.

## Figures and Tables

**Figure 1 genes-13-01739-f001:**
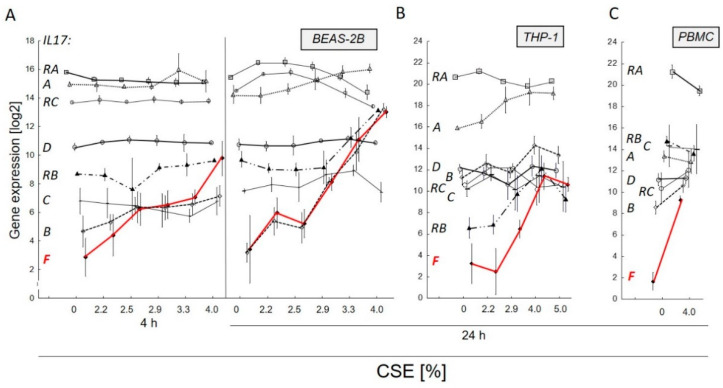
Aqueous cigarette smoke extract (CSE)-induced *IL17F*-expression in relation to IL17 cytokines *IL17A* [

], -*B* [

], -*C* [–], -*D* [

]) and IL17 receptors (*IL17RA* [

], -*RB* [

], -*RC* [

]) of members of the IL17 family in three different human cell types, epithelial lung cell line BEAS-2B (**A**), myeloid cell line THP-1 (**B**) and peripheral blood mononuclear cells (PBMC, **C**). In a concentration-dependent manner the low-copy gene *IL17F* ([

], red line) showed the strongest increase over time after 4 h or 24 h of exposure. In ordinate, the semi-quantitative frequency of transcripts to each other is considered, where *IL17F* shows the lowest basal transcript level. Gene expression was normalized to median of four reference genes (*GAPD*, *GUSB*, *PGK1* and *PPIA*). Mean and 95% confidence interval (whiskers) are depicted.

**Figure 2 genes-13-01739-f002:**
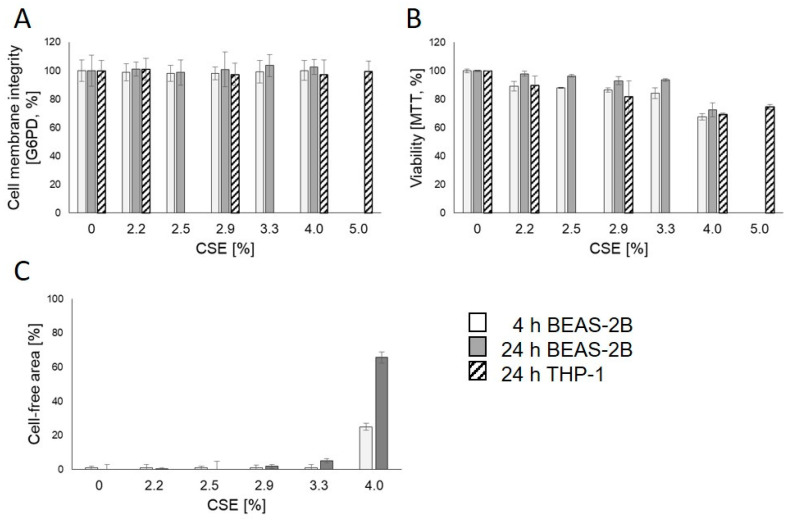
Cytotoxicity of aqueous cigarette smoke extract (CSE) in epithelial (BEAS-2B) and myeloid (THP-1) cell lines after 4 or 24 h of exposure. The cell membrane integrity was not impaired ((**A**), G6PD-assay) at any concentration whereas a strong change in cell shape from polygonal to round shape ((**C**), cell-free area) was apparent at first signs of cytotoxicity indicated by decreased viability in MTT-assay (**B**). If indicated, mean and standard deviation are depicted.

**Figure 3 genes-13-01739-f003:**
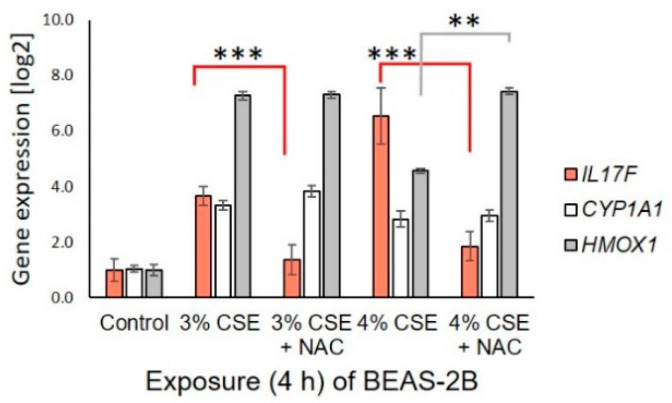
Effect of N-acetylcysteine (NAC) on aqueous cigarette smoke extract (CSE)-induced gene expression in lung epithelial cell line BEAS-2B after 4 h of exposure. NAC (0.5 mM) effectively inhibit CSE-induced *IL17F*, but not the CSE-induced AhR-mediated *CYP1A1* nor NRF2-mediated *HMOX1* expression. Gene expression was normalized to median of four reference genes (*GAPD*, *GUSB*, *PGK1* and *PPIA*). Mean and standard deviation are depicted. **, ***; *p*-value of statistical significance of *p* < 0.01 and *p* < 0.001, respectively.

**Figure 4 genes-13-01739-f004:**
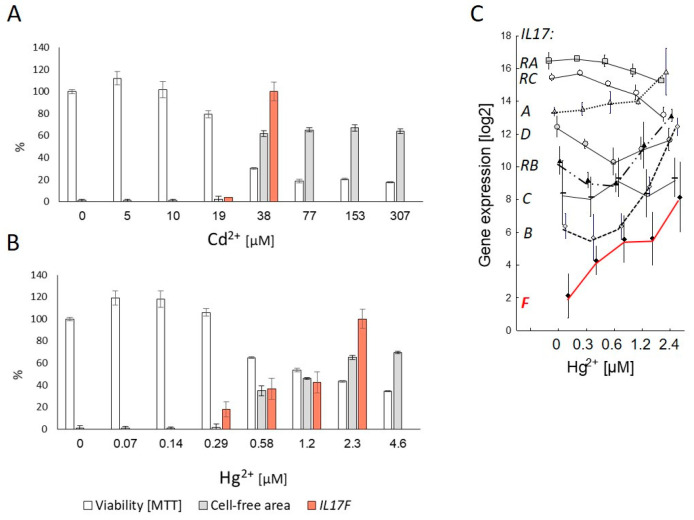
Induction of IL17F gene expression in lung epithelial cell line BEAS-2B after 24 h exposure with the two oxidative stress inducing metals cadmium (Cd^2+^, (**A**)) and mercury (Hg^2+^, (**B**)) compared to concentration-dependent cytotoxicity (decreased viability in MTT-assay) and loss of polygonal shape (cell-free area). *IL17F*-induction starts with cytotoxicity and loss of polygonal shape of adherent cells. Highest gene expression was set to 100%. The expression pattern of members of the IL17 family including *IL17F* for Hg^2+^ is similar to CSE exposure (**C**), see Figure 1. Gene expression was normalized to median of four reference genes (*GAPD*, *GUSB*, *PGK1* and *PPIA*).

**Figure 5 genes-13-01739-f005:**
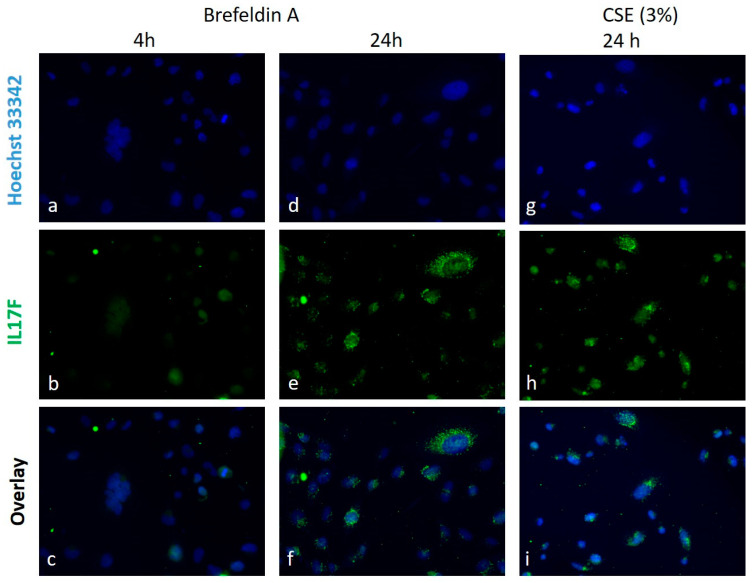
Induction of IL17F protein in lung epithelial cell line BEAS-2B by the brefeldin A and the aqueous cigarette smoke extract (CSE). IL17F staining is absent after 4 h exposure shown for brefeldin A (17.5 µM, **b**,**c**). Cytoplasmic granular pattern of IL17F was detectable for brefeldin A (**e**,**f**) and for CSE (3%, **h**,**i**) at 24 h exposure. Nuclear staining was performed with Hoechst 33,342 (**a**,**d**,**g**).

**Figure 6 genes-13-01739-f006:**
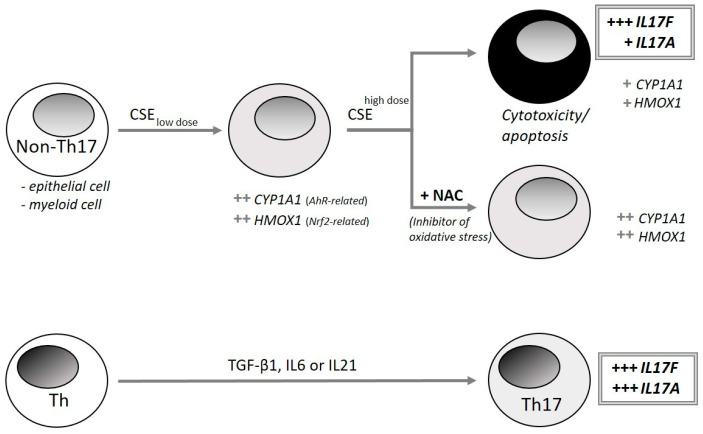
Summary of IL17F inducing conditions. In addition to IL17F from specialized Th17 immune cells [21], IL17F is also inducible by cytotoxic/apoptotic conditions in non-immune cells where the oxidative stress is decisively involved in IL17F induction.

**Figure 7 genes-13-01739-f007:**
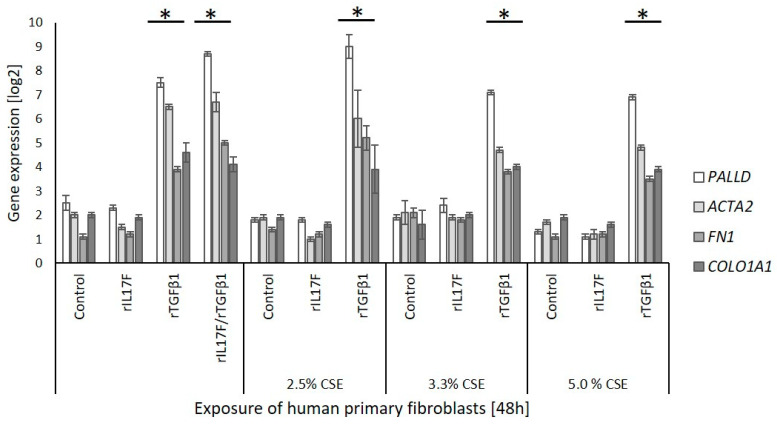
Influence of IL17F on transition of fibroblasts into myofibroblasts (FMT) examined on FMT-associated gene expression after 48 h exposure of primary human dermal fibroblasts. Fibroblasts were exposed to recombinant IL17F (rIL17F), recombinant FMT-inducer TGF-β1 (r TGF-β1) or supernatants from CSE-exposed lung epithelial cell line BEAS-2B (2.5%, 3.3%, 5.0% CSE). Gene expression was normalized to median of three reference genes (*GAPD*, *PGK1* and *PPIA*). * *p* < 0.05, paired *t*-test. FMT-associated genes: *PALLD*, palladin; *ACTA2*, α smooth muscle actin, *COL1A1*, collagen type I; *FN1*, ED-A fibronectin.

**Table 1 genes-13-01739-t001:** The family in three different human cell types (epithelial lung cell line BEAS-2B, myeloid cell line THP-1 and peripheral blood mononuclear cells PBMC) exposed to 4% aqueous cigarette smoke extract for 24 h compared to control (related to Figure 1). *IL17F* shows cell type-independent the strongest induction. Gene expression was normalized to median of four reference genes (*GAPD*, *GUSB*, *PGK1* and *PPIA*). *, ** significant differences with *p* < 0.01 or *p* < 0.001, respectively.

	Gene Expression (Fold Change)		
Gene	BEAS-2B	THP-1	PBMC
*IL17A*	3.5	10.2	−1.4
*IL17B*	1043.1 **	7.9 **	4.2 **
*IL17C*	−1.1	2.3	−5.2
*IL17D*	1.1	1.2	1.1
*IL17E*	0.9	1.1	1.3
*IL17F*	769.4 **	288.2 **	198.5 **
*IL17RA*	−2.1	−1.8	−3.5 **
*IL17RB*	10.7 **	43.7 *	−2.2
*IL17RC*	−2.1 **	−1.4	3.3 *

## Data Availability

Not applicable.

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
