# Peer review of "IL17F Expression as an Early Sign of Oxidative Stress-Induced Cytotoxicity/Apoptosis"

_genes, 2022, doi:10.3390/genes13101739_

Round 1
Reviewer 1 Report
This manuscript presented the effect of the pro-inflammatory Th17 effector cytokine IL17F in different human cell types. Overall, the paper is well written. Some specific points to address are below:
1. Presentation of Figure 1 is confusing. Some points are connected while some are not. I would suggest the authors make lines consistent and add legend.
2. Line 202 and 329, the authors made claims based on data that is “not shown”, I would suggest the authors to include all relevant data either in the main text or supplementary files.
3. Line 279-284, the text seems to be misplaced.
4. Table 4 presented aqueous cigarette smoke extract compared to control, while control values are not included in table 4. I suggest the authors to also include control value to make the comparison more straightforward to understand.
5. There are two Figure 6 (in line 272 and 379) and both are not referenced in main text.
Author Response
Overall, we want to thank the Reviewers for their time and dedication reviewing our paper. We have now revised it accordingly to their suggestions. The point-by-point reply is as follows:
Reviewer #1
Comments and Suggestions for Authors
This manuscript presented the effect of the pro-inflammatory Th17 effector cytokine IL17F in different human cell types. Overall, the paper is well written. Some specific points to address are below:
- Presentation of Figure 1 is confusing. Some points are connected while some are not. I would suggest the authors make lines consistent and add legend.
To 1.
Thanks for your comment. We are sorry that the presentation of Fig. 1 appears confusing. Lines has been indicated for all IL17 members and a legend has been included. To better distinguish IL17F its line has kept in red.
- Line 202 and 329, the authors made claims based on data that is “not shown”, I would suggest the authors to include all relevant data either in the main text or supplementary files.
To 2.
Thanks for this comment. We have now included the data formerly indicated as not shown as supplementary Figure S2 and S7.
- Line 279-284, the text seems to be misplaced.
To 3.
We apologize for this. The mentioned text has been deleted.
- Table 4 presented aqueous cigarette smoke extract compared to control, while control values are not included in table 4. I suggest the authors to also include control value to make the comparison more straightforward to understand.
To 4.
We appreciate this observation. The Table shows the data for 4% CSE from Figure 1 as fold-change gene expression compared to control what was always set to 1. Fold-change was now included in the Table legend to allow a better interpretation of the data. The link to Figure 1 has been included in table legend.
- There are two Figure 6 (in line 272 and 379) and both are not referenced in main text.
To 5.
We apologize for wrongly stating Figure 7 as Figure 6 in line 379. As proposed by a further reviewer, we moved Figure 7 into result section and re-numbered all figures accordingly.
Reviewer 2 Report
The manuscript by Bauer et al. investigates the effect of aqueous cigarette smoking extract (CSE), inorganic mercury, cadmium, and the apoptosis inducer brefeldin A on IL17F expression in different cell types (epithelial BEAS-2B, myeloid THP-1, PBMC). The authors found that CSE highly induced IL17F expression in comparison to five other members of the IL17 family (IL17A, -B, -C, -D, and -E) and three IL17 receptors (IL17RA, -RB, and RC). Further in-vitro assays showed a strong association between early IL17F induction and cytotoxicity induced by CSE, inorganic mercury, cadmium, and brefeldin A. However, a function study showed that IL17F was not involved in fibrosis. The article is interesting and the overall presentation of the article is good with appropriate use of figures. However, some of the sections do require further explanation and detail in places.
1. The basal level of IL17F expression is the lowest among all members of the IL17 family. Although IL17F was by far the most inducible gene by CSE treatment, its level remains the lowest. Why was the lowly expressed IL17F chosen as the target of this study?
2. The result of the function study is negative. The authors concluded that IL17F was not associated with fibrosis. The current study only applies to dermal fibroblasts. What about lung fibroblasts or other types of fibroblasts? Current data only support that IL17F is involved in the activation of dermal fibroblasts.
3. Real-time PCR: Did not see supplement table 1. Primer sequences are required. What is the reference gene for normalization?
4. Chapter organization: The discussion part and the conclusion part are separated.
Author Response
Reviewer #2
Comments and Suggestions for Authors
The manuscript by Bauer et al. investigates the effect of aqueous cigarette smoking extract (CSE), inorganic mercury, cadmium, and the apoptosis inducer brefeldin A on IL17F expression in different cell types (epithelial BEAS-2B, myeloid THP-1, PBMC). The authors found that CSE highly induced IL17F expression in comparison to five other members of the IL17 family (IL17A, -B, -C, -D, and -E) and three IL17 receptors (IL17RA, -RB, and RC). Further in-vitro assays showed a strong association between early IL17F induction and cytotoxicity induced by CSE, inorganic mercury, cadmium, and brefeldin A. However, a function study showed that IL17F was not involved in fibrosis. The article is interesting and the overall presentation of the article is good with appropriate use of figures. However, some of the sections do require further explanation and detail in places.
- The basal level of IL17F expression is the lowest among all members of the IL17 family. Although IL17F was by far the most inducible gene by CSE treatment, its level remains the lowest. Why was the lowly expressed IL17F chosen as the target of this study?
To 1.
We thank the Reviewer for the overall appreciation and the question that is indeed very interesting. Pilot experiments analysing the effects of different stressors on immune cell-related gene expression using the myeloid cell line THP-1 indicated a specific and strong induction of IL17F when exposed to aqueous cigarette smoke extract (CSE). Thus we aimed to find out (i) if IL17F in contrast to other members of IL17 family could serve as a marker to indicate smoking habit and (ii) what could be the trigger for this IL17F induction. Its degree of induction was for that of secondary importance. In response to the comment, we additionally implemented data from pilot experiments as supplementary Figure S2.
- The result of the function study is negative. The authors concluded that IL17F was not associated with fibrosis. The current study only applies to dermal fibroblasts. What about lung fibroblasts or other types of fibroblasts? Current data only support that IL17F is involved in the activation of dermal fibroblasts.
To 2.
Thanks for this comment. We totally agree with the Reviewer and now made clear in the revised version of the manuscript that the conclusions are more relevant for dermal fibroblasts under the current experimental conditions. It has been shown that in idiopathic pulmonary fibrosis, normal and pathogenic lung fibroblasts are inducible to myofibroblast transdifferentiation (FMT) by IL17A stimulation via functional IL17RA (Zhang et al., 2019, PMID 30604628). IL17F can bind to IL17RA or –RC and both receptors are expressed on dermal fibroblasts (Yin et al., PMID 28217129). In our work, according to Levy et al. (PMID 27930337) we confirmed an IL17RA-triggered induction of IL-6 by IL17F in dermal fibroblasts (Figure S5). We understand the limitation of our study and cannot exclude that IL17F-induced fibrotic effects may occur in other fibroblast types. We implemented this limitation in discussion section.
- Real-time PCR: Did not see supplement table 1. Primer sequences are required. What is the reference gene for normalization?
To 3.
We apologize for the fact that the supplementary Tables (S1 and S2) were not uploaded properly. They have been now uploaded, including the sequences for all genes (Table S1) which were used in our study.
- Chapter organization: The discussion part and the conclusion part are separated.
To 4.
We apologize for this. The conclusions are now included in the Discussion.
Reviewer 3 Report
The authors conduct a series of experiments to examine the induction of IL17F in response to stressors and role in fibrosis. IL17F was selected due to interest as a therapeutic target.
Below I have made suggestions on how the manuscript could be improved:
The introduction is brief and does not sufficiently bring the reader up to speed on why the research is important, the role of IL17F in various conditions and cell types is explored but the rationale for the current study is not well set up. A few statements about knowledge gaps should be included. Some of the information in the discussion should be moved to the introduction and only reiterated briefly to contextualize findings in the discussion section. For example, lines 362-365 from the discussion should be moved to the introduction, this is important information to explain the rationale for the study.
The majority of references are >5 years old. In the abstract it is mentioned that IL17F is currently of interest as a therapeutic target, more information about current work on IL17F should be included.
Figure 1 – The figure legend is not sufficient to support interpretation of the data presented. The description should be sufficient to enable the reader to interpret the table without having to refer to the text in the manuscript.
Table 4 – This table should be labelled table 1 as it is the only table in the manuscript. Units should be specified, enough description should be provided in the table legend to enable the reader to interpret the table without having to refer to the text in the manuscript.
Figure 2 – the figure is too small. Description could still be improved.
Figure 3 is also too small but the description in the figure legend is much better.
Figure 4 figure legend must be improved. Panel C of this figure is not described.
Lines 160-162. Pilot experiments with different stressors should be described briefly.
The significance of the findings and how they benefit the current state of the field needs to be better established in the discussion.
Minor: Please check abbreviations - they are continually defined multiple times throughout the manuscript even when they have been defined already previously.
Abbreviations should not be used in the abstract.
Line 279 simply says "The text continues here" - unclear what this is referring to.
There are two figures labelled figure 6. It is also unusual to introduce a figure in the discussion. I would suggest moving this to the results section.
Authors should move their discussion text from the conclusions section to the discussion section.
The first table that displays in the manuscript is labelled Table 4. This should be renamed table 1.
Author Response
Reviewer #3
Comments and Suggestions for Authors
The authors conduct a series of experiments to examine the induction of IL17F in response to stressors and role in fibrosis. IL17F was selected due to interest as a therapeutic target.
Below I have made suggestions on how the manuscript could be improved:
The introduction is brief and does not sufficiently bring the reader up to speed on why the research is important, the role of IL17F in various conditions and cell types is explored but the rationale for the current study is not well set up. A few statements about knowledge gaps should be included. Some of the information in the discussion should be moved to the introduction and only reiterated briefly to contextualize findings in the discussion section. For example, lines 362-365 from the discussion should be moved to the introduction, this is important information to explain the rationale for the study.
Response
We would like to thank the Reviewer for the overall suggestions. We have now re-worded and focussed the rationale for the study. The introduction was revised to emphasize the need for investigating conditions for induction of IL17F and the experimental conditions that we explored. The discussion was revised to emphasize the importance of our findings. As suggested, we moved lines 362-365 from the discussion to introduction and adapted the part to fibrosis in the discussion.
The majority of references are >5 years old. In the abstract it is mentioned that IL17F is currently of interest as a therapeutic target, more information about current work on IL17F should be included.
Response
Thanks for this suggestion. In the revised version of the manuscript´s introduction we now include more up-to-date references indicating the use of dual inhibitors of IL17A and IL17F for treatment of psoriatic diseases as well as ankylosing spondylitis.
Figure 1 – The figure legend is not sufficient to support interpretation of the data presented. The description should be sufficient to enable the reader to interpret the table without having to refer to the text in the manuscript.
Response
We apologized for this, we have now adapted the Figure legends to better understand the findings presented in the graphics.
Table 4 – This table should be labelled table 1 as it is the only table in the manuscript. Units should be specified, enough description should be provided in the table legend to enable the reader to interpret the table without having to refer to the text in the manuscript.
Response
Thanks for this suggestion, that was fully considered.
Figure 2 – the figure is too small. Description could still be improved.
Response
Thanks for pointing out at this. The subfigures A, B, and C are now larger and the legend was changed to better help understanding the main message of the graphic.
Figure 3 is also too small but the description in the figure legend is much better.
Thanks for this comment. We have now compiled all the data from the three subfigures into one figure and enlarged the figure.
Figure 4 figure legend must be improved. Panel C of this figure is not described.
Response
We are sorry for the poor quality of the Figure legend. We have now re-designed the figure and have add more details in the legend including panel C as needed.
Lines 160-162. Pilot experiments with different stressors should be described briefly.
Response
Thanks for this helpful comment. We have now implemented a supplementary FigureS2 to show preliminary data and have given a brief description of the pilot experiments in the results section as well as in the figure legend.
The significance of the findings and how they benefit the current state of the field needs to be better established in the discussion.
Response
We agree with the Reviewer and apologize for not having done as suggested in the first version. The discussion has been now rearranged and improved for a better understanding of the rationale of our work.
Minor: Please check abbreviations - they are continually defined multiple times throughout the manuscript even when they have been defined already previously.
Response
Thanks for this comment. We have now checked thoroughly and have deleted more than 10 repeatedly used definitions of abbreviations.
Abbreviations should not be used in the abstract.
Response
We only included abbreviations for replicated use. Thus we now indicated IL17F as interleukin 17F and deleted the abbreviations THP-1 and FMT.
Line 279 simply says "The text continues here" - unclear what this is referring to.
Response
We are very sorry for this. We have now deleted this.
There are two figures labelled figure 6. It is also unusual to introduce a figure in the discussion. I would suggest moving this to the results section.
Response
Following the suggestion by the reviewer, we introduced the figure from discussion section into results section.
Authors should move their discussion text from the conclusions section to the discussion section.
Response
We restructured this accordingly.
The first table that displays in the manuscript is labelled Table 4. This should be renamed table 1.
Response
The table has been re-numbered.
Round 2
Reviewer 2 Report
The investigators have well addressed to all the concerns raised by the referee, and they have made sufficient improvement to the manuscript.
Author Response
We thank the reviewer for his valuable comments and his final assessment that the presentation of data are now more clear and comprehensible.
The first draft was checked by a professional language service and the revised version was now checked thoroughly for additional errors. We finally made 17 additional corrections.
Reviewer 3 Report
I would like to thank the authors for their revisions which have improved the overall quality of the paper.
However, I would like to request that the authors conduct a careful check for language errors.
E.g. in the abstract:
"However, IL17F did not involve in dermal fibrosis under the studied conditions."
should read IL17F was not involved in dermal fibrosis under the studied conditions.
Line 423 in the discussion:
"We did not have examined this in detail."
should read "we have not examined this in detail".
Please note that this was not an exhaustive check of the manuscript. Other errors are likely to remain.
Author Response
We thank the reviewer for their valuable comments and their final assessment that the presentation of data are now more clear and comprehensible.
We apologize for the language errors. The first draft was checked by a professional language service and the revised version was now checked thoroughly for additional errors. We corrected the two mentioned language errors indicated by the Reviewer and made 15 other corrections.